# Validation of Low-Cost Sensors in Measuring Real-Time PM_10_ Concentrations at Two Sites in Delhi National Capital Region

**DOI:** 10.3390/s20051347

**Published:** 2020-02-29

**Authors:** Ravi Sahu, Kuldeep Kumar Dixit, Suneeti Mishra, Purushottam Kumar, Ashutosh Kumar Shukla, Ronak Sutaria, Shashi Tiwari, Sachchida Nand Tripathi

**Affiliations:** 1Department of Civil Engineering, Indian Institute of Technology Kanpur, Kanpur, Uttar Pradesh 208016, India; sahu.ravi14@gmail.com (R.S.); dixitkuldeep252@gmail.com (K.K.D.); suneeti@iitk.ac.in (S.M.); purushottamkumar1900@gmail.com (P.K.); akumars@iitk.ac.in (A.K.S.); 2Centre for Urban Science and Engineering, Indian Institute of Technology Bombay, Mumbai, Maharashtra 400076, India; rsutaria@iitb.ac.in; 3Department of Civil Engineering, Manav Rachna International Institute of Research and Studies, Faridabad, Haryana 121004, India; shashitiwari.fet@mriu.edu.in

**Keywords:** urban air pollution, PM_10_, real-time monitoring, low-cost sensors, data merging tool, data validation

## Abstract

In the present study, we assessed for the first time the performance of our custom-designed low-cost Particulate Matter (PM) monitoring devices (Atmos) in measuring PM_10_ concentrations. We examined the ambient PM_10_ levels during an intense measurement campaign at two sites in the Delhi National Capital Region (NCR), India. In this study, we validated the un-calibrated Atmos for measuring ambient PM_10_ concentrations at highly polluted monitoring sites. PM_10_ concentration from Atmos, containing laser scattering-based Plantower PM sensor, was comparable with that measured from research-grade scanning mobility particle sizers (SMPS) in combination with optical particle sizers (OPS) and aerodynamic particle sizers (APS). The un-calibrated sensors often provided accurate PM_10_ measurements, particularly in capturing real-time hourly concentrations variations. Quantile–Quantile plots (QQ-plots) for data collected during the selected deployment period showed positively skewed PM_10_ datasets. Strong Spearman’s rank-order correlations (*r_s_* = 0.64–0.83) between the studied instruments indicated the utility of low-cost Plantower PM sensors in measuring PM_10_ in the real-world context. Additionally, the heat map for weekly datasets demonstrated high *R*^2^ values, establishing the efficacy of PM sensor in PM_10_ measurement in highly polluted environmental conditions.

## 1. Introduction

One in eight deaths in India is said to be caused by air pollution, according to a report co-authored by the Indian Council of Medical Research [1]. Particulate matter (PM) includes inhalable PM (PM_10_, aerodynamic diameter ≤ 10 µm) and finer PM (PM_2.5_, aerodynamic diameter ≤ 2.5 µm), and PM-based air pollution is said to be the leading cause of deaths from ambient air pollution [2,3,4]. The annual average PM_10_ and PM_2.5_ concentrations in Indian cities were found to be 106.4 and 58.6 µg·m^−3^, respectively, with every 10 µg·m^−3^ increase in PM_2.5_ increasing all-cause mortality by between 3% and 26%, chances of childhood asthma by 16%, chances of lung cancer by 36%, and heart attacks by 44%. India, as of January 2020, has around 200 citizen-facing Continuous Ambient Air Quality Monitoring Stations (CAAQMS), which provide real-time PM_2.5_ or PM_10_ information to people. The country has over 4000 cities and towns where real-time air quality monitoring stations are required to be installed. The “affordable” or “low-cost” sensor devices to measure these PMs is a promising technology for increasing the density of the sparse urban PM pollution monitoring network [5,6,7,8]. In developing countries like India, the implementation of such technology becomes a very relevant solution for large-scale deployment of a nationwide air quality monitoring network [9,10,11]. Measurements of air pollution underpin a wide range of applications that extend from academic investigation to regulatory functions and services for the general public, governments, and businesses [12]. A nationwide dataset on air pollution is required to raise awareness of pollution and for advancing research in associated fields. Public and media attention is increasingly conscious of the health and economic expenses of high outdoor PM pollution.

Consequently, start-up companies are stepping up to produce affordable, user-friendly, and very compact wireless PM sensors to monitor air pollution [9,13,14,15,16]. These devices have the potential to bridge gaps between sparse government measurements and research groups to assess their exposure [17,18]. This wide-ranging use of this technology presumes that these portable, low-cost air pollution sensors are fit for PM measurements, although the analysis of the data quality is a subject of lively debate [19,20]. Lewis and Edwards [20] commented that the penetration of these devices into the public domain, generating a large amount of unproven data, is inevitable. Since low-cost PM sensors have not been scientifically evaluated and certified by regulatory agencies as yet, there is a significant need for benchmarking them against accurate monitors before deploying into the field conditions. Efforts from environmental technologists are of utmost importance so that these emerging technologies can realize their true potential [10,21]. The United States Environmental Protection Agency (US EPA). approved instruments for measuring PM concentrations include impactors, cyclones, Tapered Element Oscillating Microbalances (TEOM), and Beta Attenuation Monitors (BAM) [22]. These techniques are the US EPA-approved Federal Reference Methods (FRMs) of measuring PM for aerodynamic size less than 2.5 µm (PM_2.5_) and size less than 10 µm (PM_10_) from the ambient. These techniques, however, are reported to neglect the prospect of being able to correlate the variations in short-term intra-day atmospheric parameters [23,24].

Consequently, these non-continuous techniques can affect ambient particle concentration [24]. On the other hand, a continuous method could obtain PM_10_ levels through measurements by a combined system comprising of certified instruments, i.e., scanning mobility particle sizer (SMPS) and optical particle sizer (OPS) or aerodynamic particle sizer (APS). These derive particle mass concentrations from measured particle size distributions [23]. Evaluation of the SMPS–APS system has successfully determined to match well with the reference instrumentations for measuring PM concentrations [24,25,26]. Moreover, it was demonstrated experimentally that APS and OPS similarly measure PM [27]. Multi Instrument Manager Version 3.0 (MIM™ 3.0), a data merging tool, is useful in providing PM_10_ concentrations from two different research-grade equipment for measuring particulates (in two different ranges) in the absence of a single reference instrument. Notably, in emerging economies around the world, these well-proven techniques would play a vital role in validating PM_2.5_, PM_2.5-10_ (coarse particles), and PM_10_ sensors.

Continuous PM_10_ measurement instruments, including federal equivalent methods (FEMs) and other standardized research-grade devices, often cost several hundred thousand dollars and, in general, must be run in climate-controlled conditions and with extensive oversight and repairs. These instruments require significant effort to operate continuously for in-depth aerosol-driven deep statistical analysis. It is, therefore, not a cost-effective approach to rely only on these instruments to generate additional insight into pollution behavior across the country [23]. New sensor technologies may address some issues of cost and convenience posed by conventional measurement equipment. On the other hand, sensor-based PM monitors are available in roughly three orders of magnitude lower than standard instruments [10]. The overall budget of execution encompassing all other costs, such as data analytics, sensor replacement timeframe, and sensor calibration, is less well established [23]. The use of affordable miniature sensors is already underway in different applications, such as identifying hotspots for outdoor pollution, generating additional insight into pollution behavior with higher spatial and temporal resolution, mapping indoor pollution concentrations, accumulating data on personal exposure, and collecting mobile monitoring data [28,29,30]. Concerns about their precision and performance do remain unanswered [23,30].

In this manuscript, only PM_10_ is compared and not the other important metrics, such as PM_2.5_ or PM_1_. Using low-cost sensors, many authors have already well-documented PM_1_ and PM_2.5_ across the world. Another work [7] published on the performance evaluation of low-cost PM sensors by some of the authors of our research group evaluated PM_2.5_ using a low-cost PM sensor at IIT Kanpur and Duke University campuses. They validated Plantower PM sensors for PM_2.5_ measurements. They demonstrated field calibration of these PM sensors using Environmental Beta Attenuation Monitor (EBAM) as reference instruments for PM_2.5_ measurements at multiple sites with diverse environments. Mostly, the PM sensors, including the Plantower model PMS7003, are validated for PM_2.5_ measurements. However, to the best of our knowledge, no detailed study in India has been published focusing on the assessment of PM_10_ concentration using low-cost PM sensors in real-world scenarios. We compare our sensors in measuring PM_10_ levels with the co-located SMPS–OPS and SMPS–APS. We collect PM_10_ data during an intense measurement campaign period of seven weeks conducted for pollution monitoring at two selected sites in the Delhi National Capital Region (NCR), India. Merging data obtained from the certified reference instruments result in approximate overall PM mass concentrations. We validate our PM sensors for PM_10_ data with the aid of statistical tools.

## 2. Materials and Methods

### 2.1. Study Site

The two selected study sites were Manav Rachna International Institute of Research and Studies (MRIU), Faridabad, (28.45°N, 77.28°E and ~209 m above msl), and Centre for Atmospheric Sciences, Indian Institute of Technology Delhi (IITD), New Delhi, (28.54°N, 77.19°E and ~232 m above msl). These sites are in the Delhi-NCR (which is among the top-ranked polluted megacities in the world) [31]. Both the sites exist in the Indo-Gangetic Plain (IGP), which, due to its geographical components and some specific anthropogenic activities, is considered a hotspot for air pollution [32]. The selected sites suffer from heavy air pollution that masks the whole region, usually during the winter season. The field deployment of PM_2.5_ sensors at similar polluted monitoring sites is reported to perform well [7,33]. We considered only the two polluted locations in the populous Delhi-NCR, India, to cover the higher range of PM_10_ concentration. Our aim was specified to test the low-cost PM sensors in highly polluted cities and find their suitability in PM_10_ measurements.

The PM sensors were mounted on the terrace of the buildings at the respective sites. The research-grade instruments were kept inside the room, while their inlets were connected through tubing for the intake of ambient air from the outside the window. The mounted sensors and reference instruments were deployed such that their sample inlets were very close to each other to provide similar environmental conditions. At first, the two newly developed Atmos devices were installed side by side at the rooftop of Centre for Environmental Science and Engineering, Indian Institute of Technology Kanpur (IITK; 26.52°N, 80.23°E, 142 m msl), India. The ambient environment of Kanpur, India, which also comes in the IGP, is also known for its characteristic high ambient particulate concentrations. Sensors were implemented for two and a half weeks at the IITK site. After attaining sufficient data, these devices were tested for consistency, as described in the methodology section.

### 2.2. Instrumentation

New Plantower PM Sensor: The new low-cost sensors assessed in this study were Plantower PM sensors (model PMS7003). The Amos device is presented in Figure 1a. These devices are priced at only a fraction of the cost of the reference monitors. The measurement range of the PM sensors used was 0–1000 µg·m^−3^, with a resolution of ±1 µg·m³ and response time 1–10 s. The dimension of the miniature PM sensors used was 48 mm × 37 mm × 12 mm, and the temperature and relative humidity ranges were −10 °C to 60 °C and 0% to 99%, respectively. The manufacturer reported that maximum errors were relatively low (±10 µg·m^−3^ in < 100 µg·m^−3^ concentrations, and ±10% in the 100–500 µg·m^−3^ range). The detailed specifications of the PM sensor used in this study are described in Appendix A. These PM sensors use a laser-scattering technique to measure real-time PM mass concentrations and apportion laser scattering to PM_1_, PM_2.5_, and PM_10_. It is based on a proprietary algorithm that is not fully accessible by others [34]. A detailed description of the working of selected PM sensors is mentioned in a field evaluation study of PM_2.5_ by Zheng et al. [7]. We used the sensor-reported PM_10_ concentration estimates that were un-calibrated. Before field deployment, no attempt was made to calibrate these sensors under laboratory conditions due to a potentially marked discrepancy in particle size, composition, and optical properties of field and laboratory conditions [7].

Atmos—Real-Time PM Air Quality Monitors: The newly developed PM sensor Atmos box housing all the components is as shown in Figure 1b. The Atmos device uses the Plantower PMS7003 sensor for measuring PM_1_, PM_2.5_, and PM_10_ concentration values. The DHT22 sensor is used for monitoring temperature and relative humidity. The data from both these sensors are transmitted in real-time via a Quectel M66 general packet radio services (GPRS) module using 2G mobile network connectivity from local mobile service providers. A rechargeable Li-Ion battery provides backup power to the device for 10 hours. In the case of power failure, there is a seamless failover of the power module from the mains power to the backup battery. From our device itself, a local MicroSD card slot allows data to be stored and downloaded. The Atmos unit also has a Liquid Crystal Display (LCD) to view debug messages. The Atmos unit was developed in two models—with an external onboard GPS and without the GPS. The Atmos GPRS model used in this study had dimensions of length 155 mm × width 80 mm × height 60 mm. The Atmos device used the HTTP protocol to send data every 1 minute to the remote Atmos server.

Atmos—Big Data application program interface (API) and Dashboard Access: The Atmos real-time streaming data server was built using open source technologies-Apache Cassandra and KairosDB. For fast time-series database access, Apache Tomcat and HTTP server, for custom Java-based API access and HTML5/JavaScript/LeafletJS for interactive Map-based dashboard were used. Data from the Atmos device is received on the server via web services APIs and made available for comma-separated values (CSV) download and programmatic JavaScript Object Notation (JSON) access via custom-built Java APIs. Device nos. 0523 and 1292 were the two Atmos deployed at MRIU and IITD, respectively, after testing their sensitivity at ambient conditions in Kanpur, as shown in Figure 1c.

Reference Sizers: We measured particle concentration in the range of 14 nm–10 μm. We used a combination of an OPS™, Model 3330, TSI Inc., Shoreview, Minnesota, United States of America (USA) (for particles ranging from 0.3 μm to 10 μm), APS™, Model 3321, TSI Inc., USA (for particles ranging from 0.5 μm to 20 μm). The SMPS™, TSI Inc., consisting of an electrostatic classifier, Model 3082, connected to a condensation particle counter (CPC, model 3776, TSI Inc., USA (from 14 nm to 760 nm particles) as shown in Figure 1d. The time resolution for the measurements was 5 min, so that 12 data points of every hour were averaged to get hourly concentrations. The SMPS utilized a differential mobility analyzer (DMA) to classify particles as a function of electrical mobility size. At the same time, with a condensation particle counter (CPC), it determined particle concentrations, giving particle size distributions. Mass concentrations were computed through the integration of the product of the size distribution function and particle mass of every size. We acquired a continuous particle size distribution function through data inversion. It further related particulate concentration to the charging efficacy of the neutralizer, the detection efficiency of the CPC, and the transfer function of the DMA [35]. The merging process was adapted following by the method described by [36], forming the complete size distribution from 14 nm to 10 μm. However, we stated a brief on the merging process in the subsequent sub-section. We applied the necessary corrections in all the reference measuring instruments before merging and during merging, as per the requisite.

Data Merging: We merged the SMPS number distribution data with APS number distribution data using the Data Merger software Module (developed by TSI) to obtain merged mass distribution (dM/dlogDp versus Dp). During the monitoring period, we averaged the samples recorded every hour at the time of data merging. We then summed the hourly mass distributions using the trapezoidal rule to acquire hourly PM concentrations.

The SMPS number distribution data were merged with OPS number distribution data using MIM™ 3.0 (developed by TSI) in the mass mode to get PM concentrations for every hour. The MIM software is a MATLAB-based tool that allows reviewing, averaging, merging, and post-processing of data from SMPS and OPS and compiles it into a single, wide-range data set. TSI introduced it after the initial development of dedicated algorithms [37,38]. We averaged samples recorded over one hour during data merging. We then compared the PM_10_ obtained from SMPS and OPS data merging and the PM_10_ derived from the combination of SMPS and APS data. Thus, we tested merging for its suitability in getting SMPS- and OPS-acquired PM mass and also tested for over and under prediction. The merging process eliminated the discontinuity in the number distribution [39]. We took into account the inherent difference between the mobility size measured by the SMPS and aerodynamic diameter measured by the APS. During the merging, the shape factor of 1, as described by Misra et al. [40], and a widely accepted density of bulk atmospheric aerosols equal to 1.2 g.cm^−3^, was taken in the data analysis. As described by them, we combined the size distributions from SMPS and APS into a single size distribution (from 14 nm–20 µm) [40]. For this purpose, we used the TSI Aerosol Instrument Manager Program Data merge software module version 3.0.1.0.

### 2.3. Methodology

The methodology included a consistency check of the Atmos PM sensors used in similar field conditions. It also included correlating site-specific data collected from different combinations of devices and validation of Atmos PM_10_ concentrations using merged data as a reference. The schematic flowchart for this study is shown in Figure 2.

Although a single manufacturer developed the PM sensors that were used in the studied Atmos boxes, their performance in measuring PM_10_ has not yet been tested in field conditions. First, we checked the sensors for consistency, and then we deployed them in the field next to the inlets of different research-grade PM measuring instruments. We compared sensor data with the merged PM_10_ concentration from the reference instruments. The suitability of using SMPS–OPS merged PM_10_ was evaluated for reference measurements and as an alternative to well-demonstrated SMPS–APS merged PM-products [39]. However, studies have focused on getting number size distribution mostly from the SMPS–APS combination [24,26]. Simultaneously, merged PM_10_ from SMPS–OPS at both the sites (MRIU, Faridabad and IITD, New Delhi) and from SMPS–APS only at one of the monitoring sites (MRIU, Faridabad) was used to validate Atmos PM_10_ measurements.

### 2.4. Statistical Analysis

The experimental results from the deployment period were statistically analyzed using R packages, namely psych, rcompanion, and ggpubr. This included determining the mean, standard deviation, Quantile–Quantile plot (QQ-plot) formation, Pearson correlation (*r*), and Spearman’s correlation (*r_s_*). We followed a methodology as described in Ann et al. [41], Well and Myers [42], and Cohen [43]. Data were collected and arranged for analysis in a spreadsheet under Microsoft^®^ Excel^© 2020^. Furthermore, a time series for PM_2.5_, PM_10_, and coarse particles for the deployment period as well as a heat map were plotted for weekly-basis PM_10_ data during the campaign using Origin pro evaluation, 2018 software. We sought the influence of the sensors’ run-time duration on the correlation with the measured PM_10_ concentrations.

## 3. Results and Discussion

This section discusses various analyses done on the data collected during the considered deployment.

### 3.1. Consistency Test among the Sensors

Before deployment in Delhi-NCR, we co-located both the selected Atmos devices (device numbers 0523 and 1292, or Sensor 1 and Sensor 2, respectively) for a sufficient duration at IITK, India. A study conducted in Kanpur showed that PM concentration levels are quite high, similar to Delhi-NCR, with a wide-range in PM concentrations [44]. Reported studies have analyzed trends of PM_10_ in Delhi and Kanpur, India, and have found crop residue burning to be a major source. Zheng et al. [7] also selected Kanpur as a site for PM sensor field deployment with characteristic high PM concentrations. Time series and scatter plots observed for the two co-located Atmos devices (sensor one and sensor two) during this period are shown in Figure 3a,b, respectively.

The time series indicated that the ambient PM_10_ concentrations (µg·m^−3^) from two co-located Atmos PM sensors (PMS7003) were quite similar for the entire test period of two and a half weeks. We observed that the hourly time series for PM concentrations measured by both the devices matched very well. Consistency was observed between both the devices in measuring the ambient conditions with PM_10_ concentrations <100 µg·m^−3^ and ranging up to 579 µg·m^−3^. A similar time series or pattern was observed for both the devices without any ambiguities in them. In other words, no significant variation seemed to appear in measured PM concentrations (*p* < 0.05) among the two devices. For the two studied devices, for two and a half weeks, the coefficient of determination (*R*^2^) was found to be 0.97. This indicates a strong correlation between the two sensor boxes. Time series and scatter plots from the two Atmos devices were expected to be highly similar as these were from the same manufacturer, as reported by many authors [7,45]. However, due to limited field evaluation results from the manufacturer, we examined the consistency in the real-world scenario. The cause of slight variations in the ambient PM measurements may consist of instrument contamination, changed fan flow rates, and potentially inadequate cleaning of the sensors [46].

Attempting a consistency test for the PM_10_ measurement of studied Atmos sensors at MRIU and IITD sites provided high confidence in terms of their deployment in the real-world conditions or fields. Hence, these devices were then co-deployed in the Delhi-NCR next to the research-grade PM measuring instruments.

### 3.2. Time Series of Measured PM_10_ Concentrations

After collecting sufficient PM_10_ data during the deployment period of 21 January 2018–16 March 2018, time series were plotted for the two study sites, as shown in Figure 4a,b. The PM_10_ time series plots for the two sites consisted of data from SMPS–OPS, SMPS–APS, and Atmos at MRIU and data from SMPS–OPS and Atmos at IITD. Total numbers of hourly averaged data points at MRIU, Faridabad from SMPS–OPS, Atmos PM sensor, and SMPS–APS were 717, 1124, and 766, respectively. The overall mean PM_10_ concentrations measured by SMPS–OPS, Atmos PM sensor, and SMPS–APS were 98.2 ± 65.5 µg·m^−3^, 149.2 ± 86.1 µg·m^−3^, and 74.4 ± 54.6 µg·m^−3^, respectively.

The total number of hourly averaged data points at IITD, New Delhi from SMPS–OPS and the Atmos PM sensor was 832 and 1029, respectively. Mean PM_10_ concentrations measured by SMPS–OPS and Atmos were 182.3 ± 84.2 µg·m^−3^ and 181.0 ± 111.5 µg·m^−3^, respectively. The idea was to look for PM_10_ concentrations patterns of Atmos with SMPS–APS and to evaluate the correlation between SMPS–APS and SMPS–OPS, simultaneously. At both MRIU and IITD sites, trends obtained for measured PM_10_ concentrations by Atmos matched those estimated by the reference instruments. However, Atmos was generally on the higher side among the two tools measuring PM_10_ levels. Similarly, previous studies on low-cost PM sensors have shown that the sensors overestimated ambient PM_2.5_ to that with the reference monitors readings [34,47].

We measured the mean absolute error (MAE) for each of the pairs of datasets. The observed MAE value in measured PM_10_ from Atmos (uncorrected) at the IITD site concerning SMPS–OPS was 68.74 µg·m^−3^. On the other hand, observed MAEs in Atmos at MRIU site while comparing with SMPS–OPS and SMPS–APS were 56.63 µg·m^−3^ and 68.43 µg·m^−3^, respectively. These uncorrected PM_10_ readings of the sensors have an offset from that of the reference measured concentrations. The potential effect of relative humidity on particle size measurements could be attributed to the offset in measurements of PM sensors [7,17,23,48,49].

The concentration trends of SMPS–OPS were well tracked by Atmos, suggesting SMPS–OPS to be suitable for consideration as a reference for ambient PM_10_ measurements for sites with high PM_10_ concentration environments. This enhances the scope for different research communities to use affordable PM sensors and validate and calibrate PM_10_ datasets using extensive data collection by merging SMPS–OPS data.

As far as the environmental impacts are concerned, the daily PM_10_ concentrations exceeded the Indian Central Pollution Control Board limit (100 μg·m^−3^) most of the days during the studied deployment period. The PM_10_ mass concentrations at both the sites were quite high during the monitoring period of seven weeks of January-March. The PM_10_ levels are comparable to those estimated by Tiwari et al. [50] measured by the recommended method for multiple sites in Delhi in the same months. In addition to this, the measured PM_10_ mass was highest during the initial monitoring period, which gradually decreased on moving from week 1 to week 7. Nagar et al. [44] also showed that in the winter season, PM_10_ concentrations were higher during the January months in comparison to the next subsequent months. Results from our study are in agreement with the long-term research on seasonal variation and annual pattern in PM_10_ conducted in Ganga Basin [44]. Diurnal variations by both the devices matched quite well. For high PM concentrations, anthropogenic sources like biomass burning products with downwind directions were reported as one of the principal contributors. Crop residue burning is identified and well documented as the primary cause of high PM_10_ concentrations in the studied regions [51,52,53].

The total number of hourly averaged PM_2.5_ data points at IITD and New Delhi from SMPS–OPS and the Atmos PM sensor, respectively, were the same as those were recorded for PM_10_ (Appendix A). Mean PM_2.5_ concentrations measured by SMPS–OPS and Atmos were 117.31 ± 64.7 µg·m^−3^ and 161.70 ± 98.0 µg·m^−3^, respectively, for the same measurement period. Similarly, at MRIU the mean PM_2.5_ concentrations measured by SMPS–OPS, SMPS–APS, and Atmos were 65.0 ± 51.3 µg·m^−3^, 72.3 ± 52.2 µg·m^−3^, and 139.1 ± 74.7 µg·m^−3^, respectively, for the same measurement period. The PM_2.5_ concentrations from Atmos tracked well the measured concentrations of reference instruments for both the sites. Nevertheless, the values were overestimated as there was a constant offset between the measured concentrations from both.

The idea was to look for PM_10_ concentrations patterns of Atmos with SMPS–APS and to evaluate the correlation between SMPS–APS and SMPS–OPS, simultaneously. At both MRIU and IITD sites, however, Atmos was generally on the higher side among the two devices measuring PM_10_ levels. Similarly, previous studies on low-cost PM sensors have shown that the sensors overestimated ambient PM_2.5_ compared to the reference monitors readings [37,47].

Previous research works have also reported that these sensors determine size fractions differently from exact measurements of PM concentrations [54]. Similarly, the precision levels at various locations may differ depending on the chemical composition and particle size distribution [23]. Again, for the sites with a dominating source like traffic emissions, the changing size distribution on an hourly-averaged basis may also add a distinguishable change in error to the measured PM concentrations [53]. In our case as well, the MRIU site was located in the proximity of a busy cross-town roadway, which was likely to affect the performance of the different PM measuring devices. Moreover, both the locations were urban backgrounds. Hence, the respective environments could be affected by local sources such as campus vehicles, street sweeping, and other local emission sources inside the institutes. Particle measurements, when categorized across various size ranges, could be even more complicated than the analysis of gaseous pollutants. They may be altered by many parameters that vary for different measuring techniques and diverse particle kinds [55].

### 3.3. Distribution Pattern and Pairwise Correlation of Measured PM_10_ Data

At first look, the patterns of the PM_10_ concentrations from different instruments appear to track well with each other. Peaks and troughs in the measured PM_10_ levels by Atmos devices for both the sites followed those measured by reference instruments. Therefore, the observed PM_10_ time series indicated to proceed further and look for the correlations among the datasets. To select among available options to find correlation types, we analyzed using QQ-plots for the collected data from each device and from both the sites, as shown in Appendix A.

The QQ-plots, as shown in Appendix A, demonstrate the normality of data sampled from different instruments used in this deployment. The black line inside the grey area represents the normally distributed theoretical dataset, while the grey shaded area represents the theoretical confidence interval of a normally distributed dataset. The black data points represent the actual sampled data points observed for the studied instruments. If the sampled data falls within the confidence interval, it is generally assumed to be normally distributed. However, from the above figures, it was observed that the sampled data sets were above the confidence interval, which infers that the sampled dataset in our case was positively skewed. This confirms the execution of Spearman’s correlation in the collected data set.

As the data collected was positively skewed, we passed both the reference and Atmos data into the ln function, which confirmed that the data was indeed log-normally distributed. The Pearson correlation was estimated for the normally distributed data. This was done after passing the dataset to the log-function. The Pearson correlation for the normally distributed data after moving it into ln-function was 0.67 (slightly better) for SMPS–OPS and Atmos at IITD. The observed correlations between Atmos and merged PM_10_ concentrations for SMPS–APS and SMPS–OPS were 0.93 and 0.84, respectively. Similarly, in the case of PM_2.5_, the Pearson correlation after passing the dataset to the log-normal function was on the higher side. For the IIT Delhi site, it was observed that Pearson correlation = 0.94 between Atmos and SMPS–OPS. While at the MRIU site, Pearson correlation = 0.86, 0.90 between Atmos and SMPS–APS and Atmos and SMPS–OPS, respectively.

The skewed data could be explained by the high seasonal variability of PM concentrations in Delhi—January registers high PM concentrations as compared to that in March [56]. The Spearman method does not assume normality of distribution while calculating the coefficient. It is a non-parametric method of correlation, sometimes also referred to as a distribution-free test, and is often used to calculate the correlation between skewed datasets. Spearman’s correlation method has an analogy with Pearson’s method [43], which can make it comparable to the Pearson coefficient in an analysis. It was also observed that the spearman correlation method was similar when compared with Pearson’s correlation after the dataset was normalized using the ln function. The scatter plots, along with Spearman’s correlation for both the sites, are illustrated in Figure 5a–d. Pairwise correlation and data distribution of measured PM_10_ between instruments for MRIU and IITD are presented in Figure 5e,f, respectively.

We observed a *p*-value < 0.001 for all four cases; this rejects the null hypothesis and suggests that there was a strong dependence between the sampled PM_10_ measurements. The correlation is often defined as simple-specific measure and is also affected by the variability of the sampled data sets [42]. Pairwise correlations illustrated the natural distributions of different data sets collected during the study period. Correlation results revealed that there was a strong linear positive correlation among the sampled datasets. The correlation between merged PM_10_ concentrations for SMPS–APS and SMPS–OPS was found to be 0.92 and was in line with the earlier inferred results. It is hence clear from the *r_s_* values that OPS was able to capture the measurements in variation when compared with APS. Furthermore, data from the SMPS section were common in both the merged PM products. These results support the use of research-grade OPS in combination with SMPS for PM_10_ measurement purposes.

Additionally, SMPS–OPS and Atmos devices at both sites seem to appear strongly correlated. The main reason for using Spearman’s correlation was due to skewed data obtained from the instruments used for PM_10_ measurements. Pearson’s correlation method requires normal distribution as a preliminary condition to calculate an unbiased correlation. The correlation coefficient (*r_s_* ≥ 0.64) for each of the cases was observed to be quite high, which shows that there was a strong linear positive correlation. The observed findings suggest that OPS, also in combination with SMPS, acted well as the reference equipment. It may prove to be useful for developing countries’ prospects in making portable and affordable PM sensors. Periodic calibration is recommended for these low-cost sensors, as suggested by Rai et al. [29]. However, a more profound statistical attempt considering confounders’ effects on their measuring efficiency is required to ensure more confidence in such devices. The slopes and intercepts observed for the paired combinations in our study are shown in Table 1.

The pairwise correlation among sampled PM_10_ datasets also showed only 1.64 µg·m^−3^ as intercept (for *r_s_* = 0.92). These two datasets had nearly the same variation. Moreover, the parameters calculated from correlation analysis on the conducted experiments at two sites showed no significant change among them. However, local sources, including vehicular pollution nearby roads with other sources for re-suspension of dust, waste burning, several combustion sources, and secondary PM formation, are known to affect the PM-related parameters [55,57,58] routinely. Similarly, influencing factors like temperature, relative humidity, interference due to a light source, wind speed, and pressure likely bring variation in measurements.

The correlation observed between Atmos and reference instruments for measured PM_2.5_ was better than that for PM_10_ (Appendix A). Some of the recent papers from other countries have also shown that low-cost sensors’ PM_2.5_ matches more in comparison to PM_10_ [8]. In the case of PM_2.5_ as well, we observed *p* < 0.001 for all four cases. This rejects the null hypothesis and suggests that there was a strong dependence between the sampled values and PM_2.5_ measurements. Correlations ranging from results revealed that there was a strong linear positive correlation among the sampled datasets. The correlation between merged PM_2.5_ concentrations for SMPS–APS and SMPS–OPS was found to be 0.95 (it was 0.91 in the case of PM_10_) and between Atmos and research-grade instruments from 0.73 to 0.91. Since in the present study as well, the focus of the study was on the validation part of PM_10_ measurements by Atmos in measuring ambient concentration and to compare it with that of the accurate research-grade instruments, detailed information on PM_2.5_ analysis is not presented. Nevertheless, wherever necessary, we have provided the illustrations as the supplementary files.

Furthermore, coarse PM or PM_2.5–10_ (particle size between 2.5 and 10 µm) measured from Atmos devices at both the sites were compared along with the research-grade instruments at the respective sites. One time series of coarse PM for the whole duration and another time series with collocated time period with continuous period is presented in Appendix A. It was observed that the Atmos captured the coarse PM, which is comparable to the research-grade instruments, and the results were consistent for both the sites. The time series shows that the trends of the measured PM_2.5_, PM_2.5–10_, and PM_10_ from Atmos were comparable to that of the research-grade used in this study. The mean value of PM_2.5-10_ measured from Atmos was 21.39 ± 12.55 µg·m^−3^ (*n* = 702) while that from SMPS–OPS and SMPS–APS were 25.52 ± 18.74 (*n* = 940) and 9.78 ± 7.10 (*n* = 737) µg·m^−3^, respectively. Similarly, in the case of IITD site, coarse particles measured using SMPS–OPS and Atmos were 64.67 ± 33.88 and 19.75 ± 14.88 µg·m^−3^, respectively. Coarse PM data revealed that Atmos measurements underestimated the SMPS–OPS by 2.00 and 44.92 µg·m^−3^ at the MRIU and IITD sites, respectively, and overestimating the SMPS–APS by 24.80 µg·m^−3^. We also found that there was a variability in the observed coarse fraction and the ratio of PM_2.5_ to PM_10_ concentrations.

In the validation of the commodity PM sensors, a variety of reference methods were used. In our study, we did not validate our PM_10_ sensors to evaluate any reference methods. Instead, we looked for the suitability of the studied PM sensor. We also used available research-grade instruments that were considered as reference equipment to measure PM_10_. For insight into the stability of Atmos devices, the heat map for the obtained *R*^2^ was generated (Figure 6).

The *R*^2^ for each bivariate combination between the implemented PM measuring instruments for both the sites separately was determined weekly (week 1–week 7). The daily pattern of PM_2.5_ is cyclic, and in our case, the PM_10_ measurement was conducted over a single season. It indicated that weekly data should be normally distributed. We, therefore, assumed that normality would prevail after classifying the data weekly. The agreement of the Atmos (Plantower PMS7003) with the SMPS–OPS and the SMPS–APS was observed to be moderate to high for different weeks from the seven-weeks-long field deployment (*R**^2^* = 0.3–0.9). Therefore, a heat map with various shades of blue indicates the different coefficient of determinations observed from 0.3–0.9. Yellow represents the absence of data during the specific week for some of the combinations. The *R*^2^ values observed between SMPS–APS and SMPS–OPS ranged from 0.6–0.9. Clearly, the *R*^2^ between SMPS–APS and SMPS–OPS at the MRIU site was seen to be very high during the whole deployment.

The observed *R*^2^ for the data collected during the six weeks between Atmos and SMPS–OPS at IITD were 0.43, 0.52, 0.56, 0.59, 0.43, and 0.81, respectively. The number of hourly data points varied in the seven weeks time period studied. Similarly, at MRIU for the same bivariate combination, the observed *R*^2^ for consecutive weeks was 0.30, 0.45, 0.63, NA, 0.94, 0.58, and 0.44, respectively. On the other hand, with SMPS–APS as a reference for the comparative analysis of Atmos data, the observed *R*^2^ for consecutive weeks were 0.32, 0.53, 0.63, NA, 0.68, 0.60, and 0.70, respectively. Mostly, the observed *R*^2^ for the data was generally around 0.4–0.6, except for one week with a value as high as 0.9. The number of data points for PM_10_ in a week for IITD was 108–208, and that for MRIU was 22–173 with *p* < 0.001. We looked into the weekly-based data only to see the ranges of the *R*^2^ between Atmos and reference. However, there is more scope to be looked into while considering the reliability of Atmos in varying ambient PM_10_ concentration ranges. It includes the duration of data collection, the number of hourly data points, and impacting environmental parameters.

Similarly, at both the study sites, no specific diminishing concentrations were observed in the pattern of the *R*^2^ values between Atmos PM sensors and reference instruments. The results, as mentioned above for different weekly PM sensor data, showed the correlations could go very high. Furthermore, it would improve by applying the correction factors.

The difference between the numbers of collected samples for hourly averaged PM_10_ concentrations can be attributed to the varying hourly averaged data points measured during the different weeks studied. The cause of variations in the ambient PM measurements may also include instrument contamination, hardware degradation, changed fan flow rates, and potentially insufficient cleaning of the sensors [46]. Zerrath et al. [27] investigated and described that the results of the OPS matched well with the APS and SMPS. However, their investigations focused primarily on the number of concentration and modes. For the variations between SMPS–APS and SMPS–OPS, Zerrath et al. [27] also showed that equivalent diameters of urban aerosol measured by OPS and APS might differ from each other in field conditions. Variations in correlation could be due to the presence of a difference in physicochemical properties, like the density of ambient particles in different environments.

The correlation between SMPS–OPS and SMPS–APS was observed to be very high for the MRIU site. Similarly, Szymanski et al. [59] and Hand and Kreidenweis [60] also demonstrated that SMPS–APS/OPS were comparable in their experiments. Data-driven analysis indicated that SMPS–APS and SMPS–OPS are very similar in PM_10_ measurements. Furthermore, the correlation coefficients for PMS7003 sensors with both devices exhibited similar values for MRIU, which reinforces the hypothesis. No specific difference in the patterns was observed due to the change in reference data as SMPS–OPS merged data instead of the SMPS–APS combination. In developing countries like India, there are large urban areas with less or no monitoring of air pollution [6,61]. As already discussed, most of the existing pollution monitoring instruments used are expensive, hence the extensive use of low-cost sensors for PM_2.5_ and PM_10_ might be helpful in a better understanding of sources with high-resolution spatiotemporal data, along with the lesser number of monitoring stations equipped with reference-grade instruments. At the same time, big data generated from such a dense network of low-cost sensors might provide crucial information and also an opportunity for exploring further research aspects.

Additionally, seven-weeks-long data might not be sufficient to conclude the existence of drifts in the sensor measurements. For understanding the impacts of time on the performance of Atmos, a study over a longer duration of field deployment period is required. Johnson et al. [23] also mentioned that the actual response of light scattering-based PM sensors is predominantly a function of the ambient aerosol features varying with the site. Clearly, in the case of PM sensors, there is a need to explore further its size distribution and chemical composition. The envisioned better prospect of extensively available PM sensors hinges on data reliability. Hence, some of the limitations are collectively described in the subsequent section.

Limitations of the study: Like most of research studies, this study also has certain limitations, which are described below:A comparison of identical sensors generally revealed the highest agreement. Nevertheless, attempting more statistical analyses might have thrown light onto the cause of even the very slight variations among them. Accessory measurements indicating ambient temperature, humidity, and aerosol refractive index were not included in this study. The optics-based detection of particulates is probably affected by relative humidity. The uptake of moisture by hygroscopic particulates leads to increased scattered light signals. An attempt to calibrate these Atmos devices, especially for PM_10_ measurements with longer deployment duration, may help to explore more potential impacts from the variables such as relative humidity and temperature;Among the limitations of the study, lower and upper detection limits are also an expected factor in sensor performance not considered in this case. Hence, to ensure complete accuracy, the PM sensors need to be deployed in the environments where they can be tested for its performance at extreme extents. A longer duration of PM sensor deployment featuring high and low concentrations would be a challenge;Data from research-grade adjacent instruments (SMPS–APS and SMPS–OPS) were proven as suitable for PM measurements. However, to the best of our knowledge, no previous study using these instruments for similar applications is available. Hence, we suggest looking deeper into the data accuracy and uncertainties from these instruments as well as those being used as references;Transparency remained an issue with the many sensor developers where algorithms applied are valuable intellectual property. Developers and researchers should explicitly document independent algorithms to put faith in air sensor data. Hagler et al. [13] have also reported that trust in the developed sensors could augment when manufacturers would share which factors they integrated while post-processing the raw data;Likewise, most of the other available PM sensors studied Plantower PMS7003 also had no inertial-based size cuts preventing large particles from moving towards the optical chamber. It is therefore expected that it might affect the precision of readings to some extent as well. The limitations of this study also act as points to be considered as the future scope that may further serve with more information.

## 4. Conclusions

The data-driven assessment of our custom-designed sensors elucidated scopes where further advancement in its research and development can be crucial. We emphasized inter-comparison of low-cost PM sensors in the polluted sites in Delhi-NCR. The performance of PM sensors was consistent, as tested by their *R*^2^. The trends of PM_10_, PM_2.5_, and PM_2.5-10_ measurements from Atmos devices matched well with research-grade monitors. The uncorrected PM_10_ measurements by low-cost PM sensors exhibited a strong correlation with merged PM_10_ concentrations from SMPS–APS and SMPS–OPS. The Atmos devices appeared promising for PM_10_ measurement applications. Results also showed that the un-corrected PM sensors displayed consistent performance (with 0.64 ≤ *r_s_* ≤ 0.83) for PM_10_ data acquired from the research-grade instruments. During the campaign, a sufficiently high *R*^2^ value was observed between PM_10_ measured by Atmos and research-grade instruments, which also validates the sensors’ data quality.

The weekly separation of data and the regression test implies that the Atmos devices could estimate the PM_10_ level very well. In some cases, for particular instances, *R*^2^ > 0.7 was observed between the devices. The few inconsistencies where data were sufficient but the device performance was poor remain a subject of further study. Applying a calibration equation or the correction factor should improve the sensor performance for real-time ambient PM_10_ measurements.

## Figures and Tables

**Figure 1 sensors-20-01347-f001:**
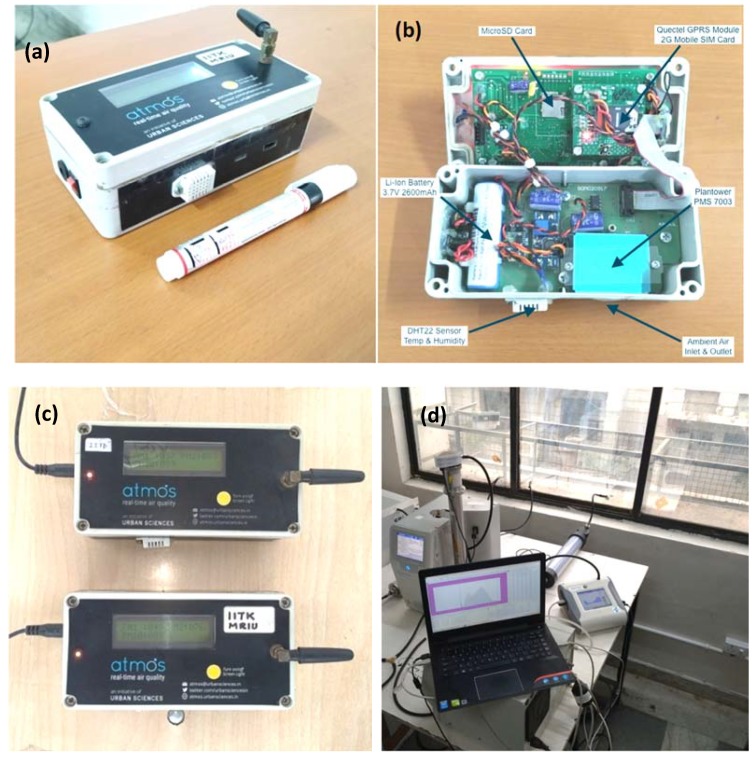
Images from the deployment of low-cost particulate matter (PM) sensors at the study site. (**a**) PM sensor Atmos comparable to the size of a marker pen, (**b**) the PM sensor box housing all components, (**c**) two sensor boxes (used in this study) co-located for consistency test, and (**d**) experimental setup at one of the monitoring sites for ambient air monitoring.

**Figure 2 sensors-20-01347-f002:**
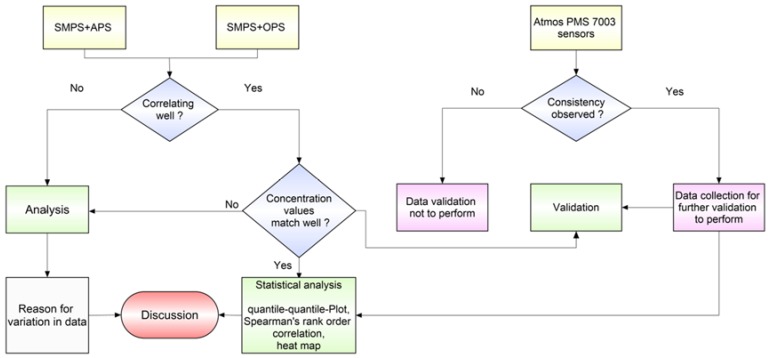
Schematic diagram showing the methodology used in this study.

**Figure 3 sensors-20-01347-f003:**
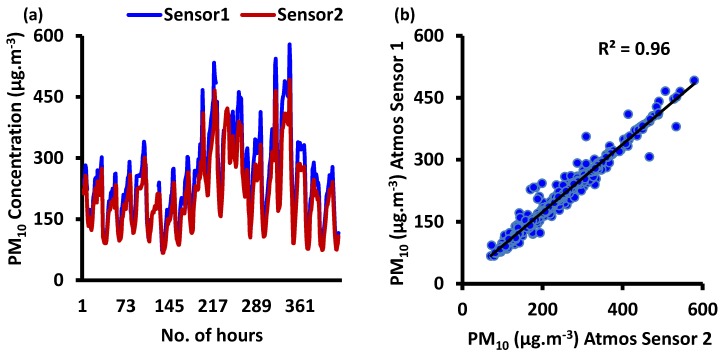
(**a**) Time-series of ambient PM_10_ concentrations (µg·m^−3^) from two co-located Atmos PM sensors for consistency test and (**b**) scatterplot for the collected data from two co-located sensors at ambient conditions.

**Figure 4 sensors-20-01347-f004:**
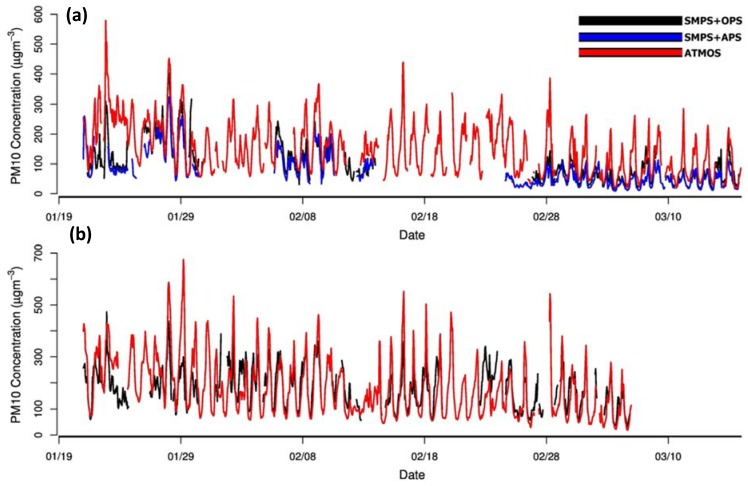
Time series of ambient PM_10_ concentrations (µg·m^−3^) data collected from Atmos PMS7003 sensor and reference instruments (merged PM_10_ from SMPS-OPS and SMPS–APS) during the deployment period at (**a**) Manav Rachna International Institute of Research and Studies, Faridabad (Delhi-NCR) and (**b**) Indian Institute of Technology Delhi, New Delhi monitoring sites.

**Figure 5 sensors-20-01347-f005:**
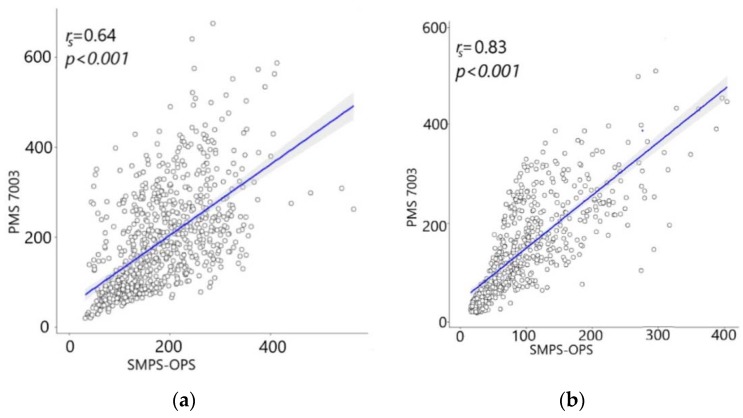
Scatter plots for measured PM_10_ between (**a**) SMPS–OPS and Atmos or PMS 7003 at the Indian Institute of Technology Delhi (IITD) site, (**b**) SMPS–OPS and PMS 7003, (**c**) SMPS–APS and SMPS–OPS, and (**d**) SMPS–APS and Atmos or PMS 7003 at the Manav Rachna International Institute of Research and Studies (MRIU) site, with their respective *r_s_* and *p*-values, (**e**) pairwise correlation and data distribution of measured PM_10_ between SMPS–OPS, SMPS–APS, and Atmos at the MRIU site and (**f**) pairwise correlation and data distribution of SMPS–OPS and Atmos at the IITD site. The grey area along the black line represents the 95% confidence interval of regression. Numeric values in upper halves represent the Spearman’s coefficients.

**Figure 6 sensors-20-01347-f006:**
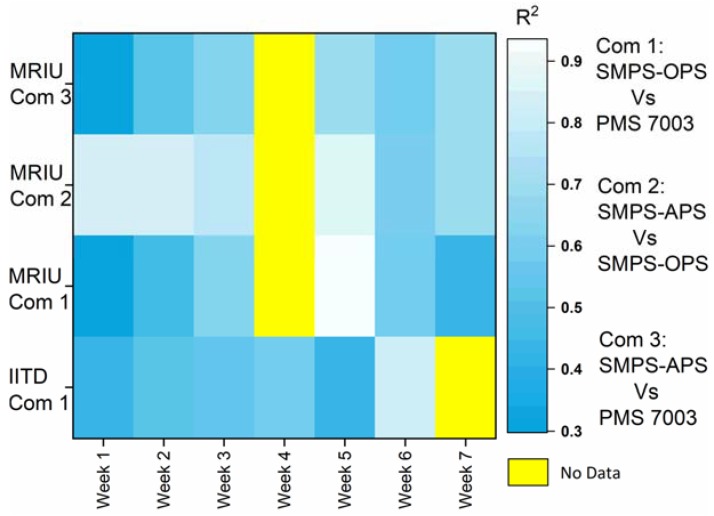
Heat map representing the coefficient of determinations (*R*^2^) for weekly basis data between the different studied instruments during their field deployment of seven weeks. The different shades of blue color represent different *R*^2^ values ranging from 0.3–0.9, and the yellow color describes that there was no data collected during the specific week.

**Table 1 sensors-20-01347-t001:** Observed parameters for different pairwise Spearman’s rank-order correlation among sampled datasets using various PM_10_ measuring instruments at two sites.

Instruments	MRIU			IITD		
	*r_s_*	Slope	Intercept (µg·m^−3^)	*r_s_*	Slope	Intercept (µg·m^−3^)
SMPS–OPS Vs. PMS7003	0.83	1.069	42.883	0.64	0.787	47.269
SMPS–APS Vs. SMPS–OPS	0.92	0.782	1.640	-	-	-
SMPS–APS Vs. PMS7003	0.83	1.188	53.396	-	-	-

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
