# Peer review of "Validation of Low-Cost Sensors in Measuring Real-Time PM10 Concentrations at Two Sites in Delhi National Capital Region"

_sensors, 2020, doi:10.3390/s20051347_

Round 1
Reviewer 1 Report
The topic presented in the article related to the use of cheap sensors for measuring air pollution is very important. This is due to the increasing air pollution in large cities. The analysis of the effectiveness of using these sensors presented by the authors can significantly contribute to effective monitoring of the air level in urban agglomerations. It will also make it easier to take measures to reduce sources of emissions of harmful exhaust components. Considering the above information, I conclude that the article is an important source of information on the low-cost sensors used to measure PM10 emissions.
Author Response
The authors would like to thank the reviewer for his positive and encouraging comments.
Reviewer 2 Report
Reviewer’s comments
Sensors
Manuscript title: Application of low-cost sensors in measuring real- time PM10 concentrations at two sites in Delhi National Capital Region
Manuscript Number: sensors-682917-peer-review-v1
General comments
This study was addressed to the feasibility of their custom-designed PM10 low-cost sensors to be replaced with the standard methods of SMPS/OPS or SMPS/APS. Their custom-designed PM10 low-cost sensors which is a laser scattering-based PM10 sensor, namely, Plantower PM sensor. A Plantower sensor is not a new and innovative technique or device to detect PM10. For validation of measurements of pecision and accuracy of PM10, parallel alignment of PM10 measurements between custom-designed PM10 low-cost sensors, SMPS/OPS, and SMPS/APS were included in the manuscript. My major concern is why the authors only put PM10 data in this manuscript if their PM2.5 data is available. For the quality of this manuscript, it is possible to be accepted for publication in Sensors after major revision based on the following comments.
Major comments
Change of the title is needed. This article is focused on validation of an analytical method of PM10 using the laser scattering-based PM10 commercial sensors. Plantower model PMS7003 is a device for detecting PM10 (maybe possible for PM2.5) in outdoor and indoor air quality. The characteristics of PM10 pollution in India particularly for Delhi and the current application or studies of detection for low-cost PM sensor. Please check and review these following articles to partially reorganize the introduction.
Chang et al. (2019) Aerosol and Air Quality Research 19(12): 2844-2864
Deshmukh et al. (2019) Aerosol and Air Quality Research 19(12): 2625-2644
Chen et al. (2019) Aerosol and Air Quality Research 19(8): 1721-1733
Do the authors know why the low-cost PM sensor studies are focused on the improvement of PM2.5, PM1.0 or more smaller PM sensor devices? The reason is for human health. I suggest the authors to review the history of PM monitoring. Why was PM10 measurement important in the past, not important in the present? Do the authors consider the interference of the temperature and humidity to Plantower model PMS7003 for measurement of PM? Please check the report of Chen et al. (2019). How about lifespan of PMS7003? Can the authors list the detailed design or instrument of PMS7003 in the supplemental materials? In figure 5, all the distribution shown in this figure is not normal. It is probable to make them as the normal distribution after transformation of logarithm scales. The correlations will be better than you used before. I suggest to use the Pearson correlation after the log-transformation.
The final question is the application of PMS7003. The results of PMS7003 shall be discuss with the other commercial devices particularly for the low-cost PM sensors with the big data application.

Author Response
The authors would like to thank the reviewer for his valuable comments. The authors have sincerely tried to address and incorporate the comments.

Reviewer 3 Report
This paper describes inter-comparison practice between established PM10 measurement techniques with low-cost, portable commercial PM sensors (Plantower). Data, analysis approaches as well conclusions were adequately described, and the writing was of sufficient quality.
I however would not recommend the publication of this paper on Sensors, for the following reasons:
There was little scientific insight. The main work done here was comparison of direct readings from different instruments, which was tedious; There was no simultaneous PM2.5 comparison. The key difference between measuring PM2.5 and PM10 is the capturing of coarse particles. Apparent good PM10 measurements can be a result of “scaling” of PM2.5 data, rather than a reflection of high-quality coarser particle measurements.
Author Response

(The authors gave the same response as above.)

Reviewer 4 Report
This document is a very interesting study of PM seniors focused on PM10 measurements. In particular the specific test field sites help rising the interest of the reader. However, the level of English writing in the abstract do not invite the reader to go ahead. That's a real pity as the document has a very interesting scientific content.
One question remains open for me, what is the interest of focusing on PM10 data form a low-costs sensors, and not including those results to a PM2.5/PM10 paper as, usually PM10 data are based on PM2.5 measurements?

Author Response
The authors would like to thank the reviewer for his valuable comments. The authors have sincerely tried to address and incorporate the reviewer's comments.

Round 2
Reviewer 2 Report
No more questions
Author Response
Response to comments of Reviewer 2
Point 1: There was no any specific comments (no more questions) but minor changes are suggested by the reviewer.
Response 1: The authors have sincerely put efforts to address all the suggestions provided by the reviewers. We have sincerely attempted to improve the scope of the manuscript from the readers' point of view. As per the suggestion, we have performed fine/minor spell check and made changes in the track-change mode. We have also added a brief on PM2.5 and coarse particle measurements from sensors.
Reviewer 4 Report
First of all, congratulations to the authors who took care to integrate the comments which have been done on there paper. I only found one mistake in line 139 where the author used µg/m3 instead of µg m-3 as for the whole document, which by the way should be written µg.m-3.
Author Response
Response to comment of Reviewer 4
Point 1: First of all, congratulations to the authors who took care to integrate the comments which have been done on there paper. I only found one mistake in line 139 where the author used µg/m3 instead of µg m-3 as for the whole document, which by the way should be written µg.m-3.
Response 1: We would like to thank the reviewer for the comment. As per the suggestion, the authors have replaced the unit used for PM concentration “µg m-3” by “µg.m-3” in the whole revised manuscript. The correct annotation for µg/m3 is incorporated as µg.m-3 at the mentioned line in the revised manuscript.
